# Application of Whole Genome Sequencing to Aid in Deciphering the Persistence Potential of *Listeria monocytogenes* in Food Production Environments

**DOI:** 10.3390/microorganisms9091856

**Published:** 2021-08-31

**Authors:** Natalia Unrath, Evonne McCabe, Guerrino Macori, Séamus Fanning

**Affiliations:** 1UCD-Centre for Food Safety, School of Public Health, Physiotherapy & Sports Science, University College Dublin, D04 N2E5 Dublin, Ireland; natalia.unrath@ucdconnect.ie (N.U.); evonne.mccabe@ucd.ie (E.M.); guerrino.macori@ucd.ie (G.M.); 2Department of Microbiology, St. Vincent’s University Hospital, D04 T6F4 Dublin, Ireland

**Keywords:** *Listeria monocytogenes*, virulence, persistence, whole genome sequencing, genotypes, biocide resistance, taxonomy

## Abstract

*Listeria monocytogenes* is the etiological agent of listeriosis, a foodborne illness associated with high hospitalizations and mortality rates. This bacterium can persist in food associated environments for years with isolates being increasingly linked to outbreaks. This review presents a discussion of genomes of *Listeria monocytogenes* which are commonly regarded as persisters within food production environments, as well as genes which are involved in mechanisms aiding this phenotype. Although criteria for the detection of persistence remain undefined, the advent of whole genome sequencing (WGS) and the development of bioinformatic tools have revolutionized the ability to find closely related strains. These advancements will facilitate the identification of mechanisms responsible for persistence among indistinguishable genomes. In turn, this will lead to improved assessments of the importance of biofilm formation, adaptation to stressful conditions and tolerance to sterilizers in relation to the persistence of this bacterium, all of which have been previously associated with this phenotype. Despite much research being published around the topic of persistence, more insights are required to further elucidate the nature of true persistence and its implications for public health.

## 1. Introduction

The genus *Listeria* belongs to the Listeriaceae family. This family belongs to the phylum Firmicutes, which is composed of Gram-positive bacteria that possess a low GC-content (36–42%) [1]. *Listeria monocytogenes* is a Gram-positive, facultatively anaerobic, non-sporulating, rod-shaped bacterium [2]. The size of these rods varies from 0.4- to 0.5-µm in diameter and from 0.5- to 2.0-µm in length. The genus *Listeria* currently consists of 21 species [3], with *L. monocytogenes* being the most significant pathogen in this genus and *L. ivanovii* being the only other pathogenic member reported [4], which rarely infects humans [5,6] but frequently causes listeriosis in ruminants [7,8]. Together with *L. marthii* [9], *L. innocua*, *L. welshimeri*, and *L. seeligeri*, these two species form the *Listeria sensu stricto* group [10], clade I, being one of two distinct clades within the genus.

All members of clade I are cultured from feces or the gastrointestinal tract (GIT) of asymptomatic animals, as well as from foods of animal origin [11,12,13,14,15], suggesting a specific interaction with mammalian hosts. Clade II, the *Listeria sensu lato* group, contains the remaining species of *Listeria* which are cultured from food-processing surfaces or the production environment. This clade includes the novel species, *L. valentina*, which was isolated from an animal farm environment in Valencia, Spain [3].

*L. monocytogenes* is a heterogenous group of bacteria. This bacterium can be differentiated into 14 different serotypes, with serovar 4 h being recently described [16,17]. These serotypes are divided into four lineages (denoted as I-IV) [18]. From the four lineages, most *L. monocytogenes* fall into lineages-I and -II [19]. Serotypes 1/2a (lineage II), 1/2b and 4b (both lineage I) cause 95% of human cases of listeriosis [20]. These categories can be further subdivided into clonal complexes (CCs), where sequence types are grouped based on their similarity to a central allelic profile [21].

Despite the low incidence of listeriosis, the mortality rate is high (20–30%) and, consequently, the burden of the disease is high [2]. This disease may manifest as a blood stream infection (BSI), meningitis, or maternal-neonatal infection following transplacental transmission [2,22]. The populations which are most at risk include the elderly, pregnant women and immunocompromised individuals. Listeriosis in humans is mostly associated with the consumption of contaminated food. Many studies reported that the contamination of food-processing facilities and associated environments is often the route by which foods become contaminated [23,24,25,26,27,28].

Cases of listeriosis are mainly sporadic. However, outbreaks have also been described [29]. Globalization of the food supply chain is likely to contribute to an increased occurrence of outbreaks. This development can be understood in terms of the wider international distribution of food. Outbreak situations may also arise due to vulnerable populations consuming contaminated food, as in the case of the documented listeriosis outbreak linked to pasteurized ice cream involving hospitalized patients who became ill [30], among other factors. The application of WGS will facilitate detection of these outbreaks. In 2019, 21 European food-borne outbreak situations were described, which was 50% higher than the previous year [31]. On an international level, the largest ever detected outbreak of listeriosis took place in South Africa from January 2017 until July 2018. During this time 1060 laboratory-confirmed cases of listeriosis were recorded, including 216 deaths [28]. Multi-locus sequence typing (MLST) revealed that 91% cases were caused by ST6, a sequence type commonly implicated in listeriosis outbreaks. The source of the outbreak was a large ready-to-eat processed meat production facility, wherein *L. monocytogenes* ST6 was isolated from environmental samples and ready-to-eat processed meat produced by the facility [28]. The advent of whole genome sequencing (WGS) has revolutionized bacterial characterization and, in turn, outbreak investigation [32,33], aiding in the global surveillance of food-borne pathogens. With many food production systems operating within a globalized supply chain, effective (global) surveillance is crucial in limiting the dissemination of food-associated microbial contaminants and protecting public health. WGS has facilitated the detection of international outbreaks of listeriosis, including an outbreak in Australia caused by contaminated rockmelons, which were linked to two cases in Singapore [34] and a US outbreak that was connected to cases in Australia, which implicated contaminated stone fruit [35].

In addition to bacterial characterization, the development of WGS technologies allows for the description of antimicrobial resistance (AMR) and virulence genes. The pathogenesis and virulence of *L. monocytogenes* is well documented [2,22,36,37,38,39,40,41,42]. Previously all *L. monocytogenes* were regarded as equally virulent. However, there is new evidence to suggest that some strains are less virulent than others, being attributed to a premature stop codon mutation in the internalin A gene, leading to the production of a truncated and inactive protein [43]. WGS is utilized to detect these truncations, allowing an assessment of the virulence potential of *L. monocytogenes*. Furthermore, hypervirulent strains which pose a greater threat to public health can also be described using WGS. These strains contain putative virulence factor-encoding genes, often mapped within gene clusters. Four virulence gene clusters have been identified, which are known as *Listeria* pathogenicity islands (LIPIs). The recognized LIPIs include LIPI-1, LIPI-2, LIPI-3 and LIPI-4, all of which have been associated with *L. monocytogenes* (Table 1). Until recently, LIPI-2 had only been associated with *L. ivanovii*. However, a partial LIPI-2 locus has been detected in atypical CC33 *L. monocytogenes* (Table 1) [42]. In a recent review by Disson, Moura and Lecuit [42], the heterogeneity in virulence is suggested to be associated with differences in host adaptation. As lineage I isolates are usually more tolerant to low pH [44], they are more likely to survive in the acidic gastric environment of a host. This observation is concordant with findings that CCs associated with hypervirulence are more effective colonizers of the gut [45]. As indicated in this review, more research is required to fully elucidate the process by which *L. monocytogenes* adapt to different environments. WGS will play an important role in this, as transcriptomic studies of different CCs are likely to clarify the phenotypes of *L. monocytogenes* in various environments.

WGS technologies have made a significant impact on the characterization of *L. monocytogenes* and have allowed for the prediction of the virulence potential of this pathogen. The remainder of this review will focus on the contribution of WGS to the assessment of the persistence of *L. monocytogenes* in food production environments and how these methodologies may advance our understanding of the persister phenotype.

## 2. Persistent Strains of *L. monocytogenes*

Persistent *L. monocytogenes* are described as isolates that are repeatedly cultured from the same source or ecological niche over time. Criteria used to define persistence have varied and this lack of a standardized definition makes it difficult to determine true persistence (Table 2). It can also be difficult to distinguish true persistence in a food production facility and reintroduction from an external source, such as a supplier of raw materials. In general, persistent *Listeria* have indistinguishable pulsotypes when repeatedly isolated from the same source at different dates over a defined period, for example, at intervals of six or more months apart [62]. Persistent strains of *L. monocytogenes* are indistinguishable when characterized using genomic epidemiological sub-typing methods. However, it is important to consider the discriminatory power of these sub-typing methods. In recent years, sub-typing methods have come a long way from the earlier multilocus enzyme electrophoresis (MEE) and restriction length polymorphism (RFLP) methods as applied by Harvey and Gilmour [63]; one of the first studies to hypothesize the persistence of *L. monocytogenes* sub-types in food production environments. Nowadays there are several molecular-based epidemiological typing methods available, including MLST and pulsed-field gel electrophoresis (PFGE), that can be applied. These methods have advantages and disadvantages which can be addressed with the application of Next Generation Sequencing (NGS) technologies which provide the best possible resolution to modern strain typing.

*L. monocytogenes* is ubiquitous in nature and can persist in food processing environments for extended periods of time. Persistent isolates may prevail for years and even decades in these food processing environments [70]. Certain phenotypes allow *L. monocytogenes* to survive and multiply in food. The bacterium’s psychrophilic nature allows it to survive and thrive at refrigeration temperatures. *L. monocytogenes* also tolerates high salt concentrations and low pH levels, which are strategies used in food preservation to limit the growth of bacterial pathogens [71]. This phenotype allows *L. monocytogenes* to contaminate a variety of foods, such as soft or hard cheese, meat and vegetables [29]. *Ready-to-eat* (RTE) foods (certain meats, fish or cheeses) are a particular cause for concern. These food products are often identified in outbreak situations as vehicles of infection [72]. RTE products do not require cooking prior to consumption and *L. monocytogenes* will survive and multiply during refrigeration. Therefore, prevention of cross-contamination within the food processing environment to food is imperative.

RTE products have been the primary focus of sampling for the presence of *L. monocytogenes.* However, other unusual sources have recently been described. Frozen corn was implicated in an EU-wide outbreak [73]. Ice-cream [74], pre-packaged caramel apples [75] and sprouts [76] were all implicated in US outbreaks. Although the number of samples tested for fruit and vegetables has increased since 2016, in 2019 less than 2% of all samples studied belonged to the fruit and vegetable categories [31]. This number is significantly lower compared to the number of RTE foods of animal origin sampled [31]. An increase in the sampling of fruit and vegetables has been suggested, as these matrices are increasingly implicated in cases of listeriosis.

Contamination by *L. monocytogenes* can also occur at the retail level. This bacterium has been isolated from environmental samples in commercial retail establishments, however, the transmission in these environments is not well understood and further studies are required. The majority of human listeriosis cases result from the consumption of contaminated RTE foods [77]. Interestingly, a higher risk is associated with products that are sliced or prepared at retail delicatessens, as they are contaminated more often and more heavily than products which do not require further processing in retail establishments.

As mentioned previously, the discriminatory power of WGS is unparalleled. Many researchers have utilized this technology in order to reduce the ambiguity of the persister phenotype, described above [78,79,80,81]. This method is useful for the detection of persistent isolates as well as outbreak investigations. Typing tools used to characterize sequence data such as cgMLST and SNP analysis can facilitate the detection of very closely related isolates that may be persisting in food production environments. Several functional concepts have been suggested to describe persistence, such as stress resistance, biofilm formation and an increased tolerance to disinfecting agents, all of which will be discussed further in the subsequent sections.

## 3. Stress Resistance

Most of the time, environmental conditions in food processing facilities are not conducive to the growth of bacteria of importance to human health. Moreover, *L. monocytogenes* in these niches, are often exposed to nutrient deficiency, heat shock, high osmolarity, low pH and competition with other microorganisms. Nonetheless, *L. monocytogenes* can adapt to a wide range of environmental stresses, including sublethal levels when later exposed to other types of stress/lethal conditions [82]. Furthermore, cross resistance to unrelated stresses resulting from previous exposure to other sublethal levels has been described [82]. Examples include resistance to peroxide following the exposure to osmotic stress [83] and resistance to heat from the exposure to low pH (Figure 1) [84]. This may be described by the vast number of mechanisms of stress resistance applied by *L. monocytogenes*, as this increases the likelihood of cross-resistance [85]. The diverse mechanisms employed by *L. monocytogenes* to survive in unfavorable conditions allows this bacterium to deploy mechanisms in response to the same stress or the same mechanism for different stresses, as is the case with heat-shock genes which are activated in response to heat stress, high hydrostatic pressure (HHP) and pulsed-electric fields (PEF) [85].

*L. monocytogenes*’ ability to survive at temperatures between −0.4 to 45 °C allows it to overcome the thermal interventional strategies that are often deployed in food preservation and production [86,87]. It is more tolerant to high temperatures when compared to other food-borne pathogens, such as *Salmonella* species and *E. coli* [88,89]. Survival at temperatures as low as −0.4 °C results in the frequent isolation of *L. monocytogenes* from refrigerated food products [90]. Another traditional method of food preservation is acidification, normally applied to dairy, meat and vegetables. The production of weak organic acids by bacteria found in the raw food or added as starter cultures have antimicrobial activity and can inhibit microbial growth through the generation of a low pH environment [91,92]. *L. monocytogenes* exhibits an adaptive acid tolerance response (ATR) whereby exposure to mild acid stress (pH 5.0) leads to tolerance to subsequent exposures at an even lower pH (3.0) [93,94]. Bacteria experience osmotic stress in the presence of increased concentrations of salt or sugar, humectants that are added to foods such as cheese, seafood, pickles, jams and syrups as preserving agents [95]. The concentrations of these additives alter the osmotic balance between the cytoplasm of a bacterium and its external environment [96]. Furthermore, *L. monocytogenes* can reduce osmotic pressure and water loss through the accumulation of osmolytes within the cytoplasm in the presence of high salt concentrations [97]. Nisin a bacteriocin with a broad activity against Gram-positive bacteria can be used as a preserving agent [98]. This antibiotic has two modes of action against bacteria; inhibition of cell wall synthesis and formation of pores in the cell membrane, resulting in permeabilization [99]. Not surprisingly, *L. monocytogenes* has developed resistance to nisin, mediated through changes in the cell membrane composition [100,101].

A recent study by Hingston et al. [44] revealed that *L. monocytogenes’* survival in stressful conditions can be linked to its serotype, clonal complex, full length *inlA* profile and plasmid harborage. In this latter study, a full length *inlA* gene contributed to an increased tolerance to cold temperatures, while the presence of plasmids resulted in an increased acid tolerance. There are two phylogenetic groups of *L. monocytogenes* plasmids which are distinguished by their *repA* sequences [102]. Although plasmid harborage can result in a reduced bacterial growth rate due to increased metabolic demand [103], certain genes mapped on plasmids can, in contrast, provide *L. monocytogenes* with a growth advantage. Genes mapped on *L. monocytogenes* plasmids include benzalkonium chloride resistance genes (encoded by *bcrABC*) [104,105,106] and cadmium resistance genes (encoded by *cadA2, cadAC*) [105,106,107]. Oxidative stress response genes such as peroxidases and reductases have also been found to be located on these plasmids [102]. More research is needed to elucidate the function of other genes also located on *L. monocytogenes* plasmids and how they contribute to stress tolerance (i.e., acid and salt) and persistence. As a full length *inlA* gene was more prevalent among isolates that are tolerant to cold, salt and acid, it can be suggested that the presence of surface proteins such as InlA can contribute to protection against certain stresses. In contrast to this, a study by Franciosa et al. [108], found that truncations in *inlA* contributed to increased biofilm formation, while a study by Piercey et al. [109] reported isolates with truncations in *inlA, inlB* and *inlH* to be better biofilm formers. These results suggest that internalins may have a function in processes other than virulence.

*Listeria* stress survival islands include SSI-1 and SSI-2 (Figure 2) (Table 3). SSI-1 contains 5 genes which contribute to *L. monocytogenes* tolerance to cold, acid and salt [110]. Some studies have found no significant variations in responses to cold, acid or salt between isolates which contain SSI-1 and those which do not [44,111,112]. However, this may be the case when comparing large numbers of isolates, rather than a mutant and its isogenic wildtype. A study by Ryan, Begley [110] which compared a SSI-1 harboring strain with its isogenic mutant reported that loss of SSI-1 significantly compromised *L. monocytogenes*’ ability to grow and survive in frankfurters, which have previously been implicated in outbreaks of listeriosis [113,114,115], as well as growth at 4- and 15-degrees C. These data, along with the absence of SSI-1 from the genomes of *L. monocytogenes* which are not commonly associated with foods, suggest that genes found in SSI-1 may be involved in the bacteriums’ ability to survive in food [116,117,118,119,120]. SSI-1 is located in a hypervariable *hot spot*, wherein homologues of *L. innocua* genes *lin0464* and *lin0465* can also be found [121]. It has been demonstrated that these genes can also be found in strains which belong to sequence type (ST) 121 [1,121,122]. These strains are often found to be persisting in food processing environments [123,124]. Therefore, it is possible that these loci have a function in adaptation to stress.

A more recent study by Harter, Wagner [127] investigated the role of these stress islands by creating a *lin0465* deletion mutant using a persistent *L. monocytogenes* belonging to ST121 which was repeatedly isolated from an Irish cheese production facility over a 12-year period [128]. The isolate was exposed to a variety of stress conditions, including acid, gastric juice, cold, heat, osmotic, alkaline, oxidative and antibiotic stresses. This study characterized this gene insertion as SSI-2 and phenotypic characterization of strains possessing SSI-2 have conferred resistance to stresses different to those encoded by genes mapped to SSI-1 (Figure 2) (Table 3). SSI-2 is believed to be involved in the response to oxidative stress which occurs as a result of a high concentration of oxygen radicals leading to cell death due to irreversible damage to proteins, lipids and nucleic acids [129]. The resistance of *L. monocytogenes* to oxidative stress is suggested to be associated with biofilm formation [129]. Harter and Wagner [127] also reported that, unlike SSI-1, the regulation of SSI-2 is independent of the alternative stress sigma factor (*sigB*). Although SSI-2 does not confer resistance to QACs, the novel transposon Tn*6188* involved in QAC resistance is harbored by ST121 strains [122,130,131].

It has also been shown that although isolates are very closely related and belong to the same clonal complex, they can still exhibit significant differences in stress tolerance patterns [44,132]. These variations can be attributed to small genetic differences which result in a significant effect on the phenotype. Small genetic variations can be detected through genome-wide association studies (GWAS). Although these studies have been hugely effective, they can be of limited value when it comes to bacterial genomes due to the high frequency of mutations encountered [44,133,134]. Hingston, Chen [44] applied GWAS to their collection of *Listeria* in order to detect SNVs responsible for stress tolerance. Their results showed no common SNVs in the stress tolerant isolates. However, unique SNVs were recorded in up to four stress susceptible isolates. The gene *sigB* facilitates *L. monocytogenes* to survive in unfavorable conditions and this has been well documented [99,135]. Notably, common SNVs found among the collection of stress sensitive isolates resulted in PMSCs in several regulatory networks controlled by *sigB*. These included PMSCs in *rsbS* and *rsbV* [136] in the cold sensitive group; PMSCs in *rsbS* among three desiccation sensitive isolates; and PMSCs in *rsbU*, a post transcriptional regulator of *sigB* in two other desiccation sensitive isolates. In another GWAS study reported by Fritsch et al., [137] a slow growth phenotype was established among the isolates when grown at two degrees for two months. A SNP based GWAS revealed 184 significant SNPs and a gene based GWAS found 114 significant associated genes. Some of these SNPs/genes were known to be associated with *L. monocytogenes’* tolerance of cold temperatures and others were linked to hypothetical proteins. There have been few GWAS studies performed to assess phenotypes relevant to food safety, however these methods have the potential to further our understanding of the persistence of *L. monocytogenes* in food environments.

The continuous emergence of stress resistance among isolates of *L. monocytogenes* cultured from food matrices and/or their associated production environments led to the establishment of novel methods for food preservation. These include high pressure processing (HPP), the application of UV-light, pulsed-electric field processing and oxidative stress. *L. monocytogenes* have also evolved to adapt to these novel technologies. HPP is an alternative processing technology to thermal treatment and is designed to increase the permeability of the bacterial cell membrane by disrupting structural proteins, therein ultimately leading to the inhibition of metabolism, replication and transcription [138]. Resistance to HPP, expressed by *L. monocytogenes*, appears to be strain specific [139]. Pulsed-electric field processing is also an alternative to thermal treatment. This strategy is mostly used in the processing of liquid foods, and it acts to irreversibly damage the cell membrane, leading to leakage of cytoplasmic contents [140]. Gram-positive bacteria are more resistant to this treatment when compared to Gram-negative bacteria; an observation which may be attributed to their thicker cell wall [141]. Exposure to UV-light eradicates surface microorganisms that have contaminated food products during processing stages [142]. UV-light acts on bacteria in a number of ways, including by damaging DNA along with exerting photophysical and photothermal effects which lead to leakage of cellular contents [142,143]. Although specific *L. monocytogenes* mechanisms of resistance to UV-light are unknown, it has been demonstrated to be more resistant to UV-light than other pathogens such as *E. coli* [144].

## 4. Biofilm Formation

One of the main sources of repeated contaminations in food production is the growth of biofilms [145,146,147]. Biofilms are an accumulation of bacterial cells, encased in a self-produced exopolysaccharide matrix which have an enhanced ability to attach to surfaces [148]. Bacteria enclosed within this matrix are protected from stresses arising from the external environment, such as desiccation, nutrient deprivation and disinfection procedures [149,150]. Previous research findings into biofilm formation by *L. monocytogenes* have been inconsistent and contradictory. However, many recent studies have shown that this is a significant strategy deployed for the survival and persistence of some *L. monocytogenes* [147,151,152,153]. *L. monocytogenes* can produce biofilms on a variety of materials found in food production facilities, such as buna-N rubber, cast iron, stainless steel, nylon, teflon, polyester floor sealant and glass [154,155,156,157].

Earlier studies explored the relationship between biofilms and the persistence of *L. monocytogenes*. Some authors reported persistent strains to be better biofilm formers when compared to sporadic strains. Manso and Melero [158] studied biofilm formation by *L. monocytogenes* on plastic surfaces and on stainless steel. Here it was found that *L. monocytogenes* only formed biofilms on polyvinyl chloride (PVC), however, the strongest biofilm formers were identified as persistent strains. Lee and Cole [159] found no association between persistence and biofilm formation. Other researchers investigating the ability of *L. monocytogenes* to form biofilms utilized a simple microtiter plate method, followed by crystal violet staining, as described by Stepanovic and Cirkovic [160] (Table 4). Using this approach, researchers reported *L. monocytogenes* to be weak/intermediate biofilm formers [158,161,162]. A major disadvantage of this method is that crystal violet stains both the bacterial cells and the extracellular matrix and, thus, only measures the biofilm mass, not its viability. Metabolic assays have been suggested to overcome this limitation, as they quantify the viability of the biofilm itself. An example of this is the use of the resazurin assay, wherein a blue dye is reduced to pink resorufin by metabolically active cells, embedded in the biofilm.

The relationship between biofilm formation and serotype/lineage has been commented upon, but with conflicting results. Takahashi and Miya [163] observed significantly higher biofilm-forming ability among isolates of lineage I using the CV assay. Likewise, Djordjevic and Wiedmann [166] reported a higher biofilm-forming ability for lineage I (serovars l/2b and 4b) than for lineages-II (serovars 1/2a and l/2c) and -III, using the same method. Contradictory to these findings, Borucki and Peppin [164] found a higher biofilm-forming ability for isolates of lineage II. Barbosa and Borges [165] compared the biofilm forming ability of food and clinical isolates, and reported a significant difference. The majority of clinical isolates in this study were weak biofilm formers, while 40% of food isolates were moderate formers. Serogroups IIa and IIb (and IIc for food isolates) included the highest percentage of isolates showing the strongest ability to form biofilms at 37 °C. This study also compared biofilm formation at 4- and 37-degrees C, with increased biofilm formation being recorded at the latter temperature. It has also been suggested that serotype-specific differences in biofilm formation patterns of *L. monocytogenes* can be linked to the presence of SSI-1 [167]. In a study by Keeney and Trmcic [167], strains of serotype 1/2b were the strongest biofilm formers; the majority of which contained SSI-1. The weakest biofilm formers belonged to serotype 4b, the majority of which did not harbor SSI-1. The conflicting results in investigations of biofilm formation being related to persistence could also be a result of the varying criteria used to determine persistence or the discriminatory power of the methods used to detect persistence. In addition, the serotypes of these presumptive persistent strains varied across studies. As different serotypes are associated with infection and the food environment, they may be adapted to form biofilms at different temperatures, (i.e., clinical strains favoring the body temperature and food strains favoring lower temperatures). Another reason for conflicting results in studies attempting to explore the association of the biofilm phenotype with lineage, serotypes and origin may be caused by the differences in methods applied and the strains selected for study.

As the majority of attempts to link biofilm formation to serotype or persistence have been conducted using phenotypic investigations, genomic methods may have the potential to overcome these conflicting results. An example of this is a study by Lee and Cole [159] who utilized GWAS, wherein large numbers of genetic variants were tested with a given phenotype. Although there were definite differences in the biofilm formation by strains belonging to different serotypes, they were inconsistent when all the growth conditions were taken into account. In fact, associations based on lineage, yielded conflicting results, as two serogroups (IIb and IVb) belonging to lineage I presented significant variations in biofilm formation under some selected conditions. These authors suggest that instead of using these criteria to determine the intraspecific differences in biofilm formation, other genetic markers, such as strain types or clonal complexes, may prove to be more reliable. In this study, CC26 produced more biofilm at 10 degrees C, which indicates that core genes are involved in survival under cold temperatures. This hypothesis has also been explored by other authors. For example, Maury and Bracq-Dieye [45] recorded that the genomes of certain clonal complexes may present advantages when exposed to environmental stress, as they have shown that hypovirulent clonal complexes (CC9 and CC121) were better biofilm formers when exposed to sub-inhibitory concentrations of benzalkonium chloride. Hingston and Chen [44] reported similar associations in relation to stress tolerance traits. Most studies implementing GWAS focused on clinical phenotypes. Nevertheless, GWAS is emerging as a tool that can be applied to address food safety questions—as in the case of investigating *L. monocytogenes* traits associated with cold, salt, acid and desiccation stresses [44,137].

To date, many genetic determinants have been linked to biofilm formation in *L. monocytogenes* a feature that demonstrates the complexity of the phenotype being studied. Transposon mutagenesis is a method often utilized to investigate the functionality of genetic determinants responsible for biofilm formation and several studies focused on examining the effect of knockouts of individual genes involved in biofilm formation. The genes involved in biofilm formation include the *degU* (virulence) orphan response [168] and *flaA* [169] both of which control motility [170,171]. The *agrBDCA* operon is involved in quorum sensing [172,173]. The *L. monocytogenes* Biofilm-Associated Protein (BapL) contributes to the surface adherence of strains that possess it, however it is not essential for all strains [174]. Similarly, the transcriptional regulator of stress response genes, SigB, has been shown to exert an effect on biofilm formation by van der Veen and Abee [175]. Although these studies are useful, identification of functionally relevant biofilm genes on a genome-wide level would be of value. Chang, Gu [176] conducted one of the first GWAS studies using the mariner-based transposon mutagenesis approach. This study identified 24 genes involved in biofilm formation. Alonso and Perry [177] identified 38 genetic loci believed to be involved in *L. monocytogenes* biofilm formation when grown at 35 degrees C. Among these loci were the D-alanylation pathway genes *dltABCD* and the phosphate-sensing two component system *phoPR*, the significance of which was explored by the creation of deletion mutants. The result of this was a significant reduction in biofilm formation by the mutants. These results are indicative of the significance of the D-alanylation of lipoteichoic acids mediated by the gene products of the *dltABCD* operon and the phosphate-sensing *phoPR* two-component system in *L. monocytogenes*’ ability to form biofilms. A list of genes involved in biofilm formation by *L. monocytogenes* is found in Table 5.

A study by Piercey and Hingston [109] identified genes that are involved in biofilm formation at 15 degrees C, a temperature more commonly encountered in food production environments. It is likely that these genes may differ, as the environmental temperature affects cell surface hydrophobicity [178], motility [179] and the expression of other genetic factors [180,181]. This study also utilized transposon-mediated mutagenesis to create mutant libraries. Nine *loci* associated with biofilm formation were identified that were not previously described to be linked to biofilm formation at higher temperatures. A Pan Genome Wide Association Study conducted by Lee and Cole [159] compared sets of genes identified to be related to biofilm formation at 37- and at 10-degrees C and which included four different conditions. The conditions tested included BHI, distilled BHI, BHI + NaCl and distilled BHI + NaCl. At 37-degrees C some 1360 genes were identified with only 50 genes (3.68%) found in three or more conditions. At 10-degrees C, some 1050 genes were identified, with 59 genes (5.62%) common in more than three conditions. The products of the identified genes were found to encode cell surface proteins and transformation/competence-related functions.

To fully elucidate the process of biofilm formation, further genetic investigations are required such as examining unknown gene functions and transcriptomics/proteomics approaches. Furthermore, it would be interesting to explore the effects of the relationship between biofilm formation and biocide resistance on the persistence of *L. monocytogenes* in food production environments (FPEs), which is discussed further in the following section.

## 5. Biocide Susceptibility

Susceptibility to biocides used to eradicate *L. monocytogenes* has been described as a potential mechanism for persistence due to the survival of certain strains post-cleaning/disinfecting procedures. This tolerance may be caused by the presence of niches within food processing environments which are inaccessible to disinfectants, resistance to biocides arising from the acquisition of genetic markers, and/or formation of biofilms which inhibit the penetration of biocides. Biocides are extensively applied in food production environments to prevent bacterial contamination, and common examples include: quaternary ammonium compounds (QACs), aldehydes, alcohols, chlorine-releasing compounds and peracids. The modes of action of biocides are similar to that of antimicrobial agents, involving targets such as cellular constituents, cell membranes and thiol groups [182]. Resistance to biocides among *L. monocytogenes* has been recognized for decades and is on the rise, posing a challenge in the elimination of *L. monocytogenes* from the food processing environment [183,184]. The extensive use of biocides and exposure to sub-inhibitory concentrations can create a selective pressure which can then cause resistance through mutation or horizontal gene transfer [182]. Sub-inhibitory concentrations of biocides in the environment may stem from inadequate cleaning prior to disinfection, disinfection of wet surfaces, and incorrect dosing [185,186].

Benzalkonium chloride (BC) is a common QAC applied in food production which has a broad spectrum of activity [187]. BC targets cytoplasmic membrane permeability functions [182]. Reduced susceptibility to BC, expressed by some isolates of *L. monocytogenes*, has been documented since 1998 and several genetic determinants have been linked to this phenotype. In particular, small multidrug resistance (SMR) protein family-encoding genes are associated with reduced susceptibility to BC. In *L. monocytogenes* these genes include *qacH* mapped to the transposon Tn*6188* [122], *emrE* located on the genomic island LGI1 [188], *emrC* [182] and the *bcrABC* cassette [104]. This latter cassette consists of a transcriptional regulator gene (*bcrA*) along with two SMR efflux pump genes (*bcrB* and *bcrC*) [104,189]. Along with SMR efflux pumps, major facilitator superfamily (MFS) efflux pump-encoding genes such as *mdrL* and *lde* have also been described in *L. monocytogenes* [190,191,192]. These pumps are acquired from recombinant elements and mobile genetic elements (MGE) through horizontal gene transfer. The acquisition of resistance mechanisms to QACs can result in persistence of *L. monocytogenes* in FPEs, as the concentration of remaining QAC may be inhibitory for susceptible strains, yet subinhibitory for resistant strains, as demonstrated by Ortiz and Lopez [193]. In their study, certain resistant strains of *L. monocytogenes* were able to form biofilms at 5 mg L^−1^ BAC, an inhibitory concentration for the majority of strains.

Many studies reported the presence of *L. monocytogenes* resistant to QACs in different food processing facilities [65,124,194,195]. Several have also linked QAC resistance to persistence of *L. monocytogenes* in food production environments [65,195,196,197]. BC resistance in *L. monocytogenes* is considered low-level and is unlikely to lead to resistance to the concentrations of these compounds usually applied in the food industry (generally 200–1000 mg L^−1^) [70,194]. This low-level resistance, expressed by these efflux pumps, can lead to the reduction of intracellular biocides to sub-inhibitory levels, leading to prolonged survival [198]. The correlations between low-level resistance to QACs and persistence of *L. monocytogenes* in FPEs are scarce [124,193,199,200,201]. This can also occur in the environment. In domestic wastewater, for example, the average concentration of QACs is around 0.5 mg L^−1^, decreasing to 0.05 mg L^−1^ in wastewater treatment plant effluents [194]. This can lead to environmental persistence of *L. monocytogenes* which express low-level resistance to QACs.

However, it is often a challenge to interpret data describing the biocide resistance of bacteria. This is due to the lack of standardized criteria defining resistance to disinfectants and, often, *L. monocytogenes* isolates are characterized as resistant based on survival at concentrations which inhibit most strains [202]. There is also a need for standardized protocols to study biocide susceptibility, as MICs may vary when different methods are applied. Additional investigations are also required into biocide resistance patterns of planktonic versus sessile cells, as the increased resistance expressed by biofilms to biocides is believed to be reduced once bacterial cells return to their planktonic state [149,203]. The ability to form biofilms can vary among strains which are equally resistant to BAC, but have different genetic determinants of BC resistance. This is evident in a study by Ortiz and Lopez-Alonso [201] wherein *L. monocytogenes* belonging to ST121, containing the gene *qacH*, formed biofilms at 5 mg L^−1^. Meanwhile s ST31 isolate which was also resistant, formed no biofilm at the same concentration. Nakamura and Takakura [204] have also shown that QAC resistant *L. monocytogenes* are quicker to form biofilms than sensitive strains. It is possible that the selection of resistant *L. monocytogenes* due to exposure to subinhibitory concentrations of QACs may allow for biofilm formation and subsequent persistence in FPEs. As BC tolerance and stress resistance genes are often found on MGEs, susceptible isolates can readily become resistant in the presence of resistant isolates [102,104,130,205,206]. To combat the spread of BC resistance genes and subsequent persistence in FPEs, the use of non-chemical or combined antimicrobial approaches has been suggested [207,208,209]. CC1 and CC4 are commonly associated with infection, while CC9 and CC121 are related to food [61]. CC2 and CC6 that usually harbor BC resistance-encoding genes are associated with infection to a lesser extent than CC1 and CC4 [45]. However, a recent outbreak in Switzerland was attributed to *L. monocytogenes* CC6 that was found to be persisting in a cheese production facility [210]. Furthermore, *L. monocytogenes* CC6 is progressively implicated in outbreaks, such as the European-wide outbreak linked to five countries between 2015 and 2018 [73], an outbreak in Switzerland in 2016 [208], the largest global outbreak which occurred between 2017 and 2018 in South Africa [211] and the largest European outbreak in the last 25 years that occurred in Germany [212]. These results may indicate that virulent strains could be simultaneously resistant and persistent within food production environments, having detrimental effects on the food production industry and public health.

## 6. Conclusions

Many studies have failed to clarify the association between *L. monocytogenes* persistence and biofilm formation and biocide resistance. As a result, it has been argued that long-term persistence of *L. monocytogenes* may simply be due to the survival and growth in specific ecological niches found within the food processing environment, rather than adaptation of the bacterium itself. These specific niches may include cracks and crevices of surfaces along with seals and gaskets that can be difficult to clean and disinfect. However, with the advent of novel genome characterization methods, better comparisons among isolates of *L. monocytogenes* can now be made. These strategies have allowed researchers to determine the specific distribution of *L. monocytogenes* CCs identified in different environments, as well as to discover unique characteristics associated with them. Epidemiological links between persistence, resistance to disinfectants and biofilm formation have been made in numerous studies, however evolution studies of biofilm formation in response to subinhibitory levels of disinfectants could prove more insightful to describe this relationship. With the emergence of standardized protocols for the measurement of biocide MICs, it is likely that results among different studies will now be more concordant. With the continuous advancement of molecular- and phenotypic-based methods, the ambiguity surrounding our understanding the persistent nature of *L. monocytogenes* in food production environments will be clarified.

## Figures and Tables

**Figure 1 microorganisms-09-01856-f001:**
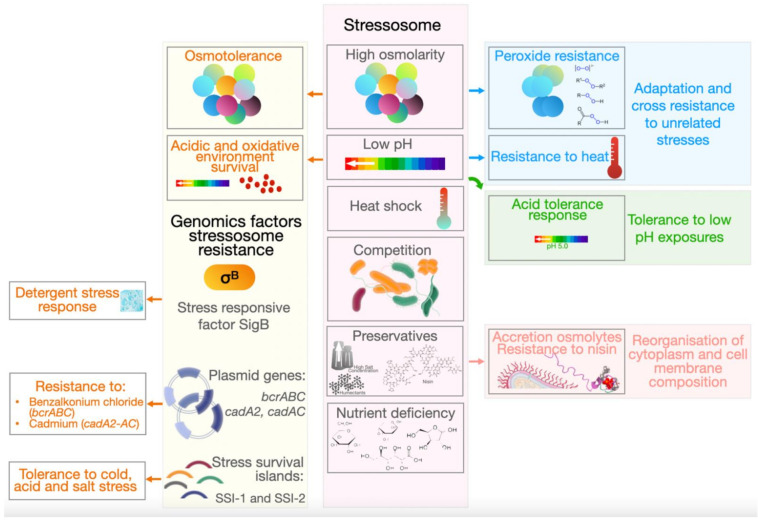
*L. monocytogenes* adaptation to stress conditions (stressosome) involves various genomic factors and causes different adaption phenomena.

**Figure 2 microorganisms-09-01856-f002:**
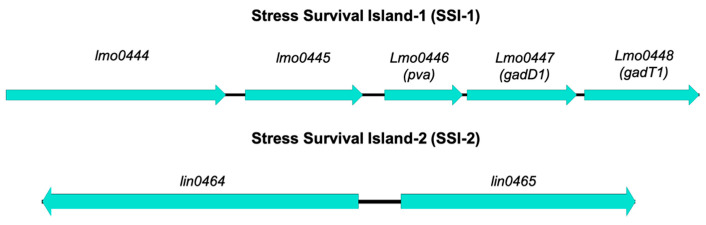
Scheme of the genetic organization of genes mapping to stress survival islands. Created using Easyfig [33]. (see also Table 2).

**Table 1 microorganisms-09-01856-t001:** Pathogenicity islands found in *Listeria monocytogenes*.

Island	Gene	Product	Function
LIPI-1	*actA*	Actin assembly-inducing protein	(i) Actin recruitment and polymerization events responsible for intracellular movement and cell-to-cell spread [46] (ii) autophagy evasion [47] (iii) Mediation of aggregation and biofilm formation [48] (iv) May also be involved in the entry into eukaryotic cells, probably by recognition of an HSPG receptor
	*mpI*	Metalloprotease	Process the PC-PLC into its mature and active form [49]
	*plcA*	Phosphatidylinositol-specific Phospholipase C	(i) Hydrolyze glycosyl PI (GPI)-anchored eukaryotic membrane proteins [50] and (ii) synergize with LLO and PC-PLC in the destabilization of primary and secondary phagosomes [51]
	*plcB*	phosphatidylcholine-specific Phospholipase C or lecithinase	(i) Synergize with LLO and PI-PLC in the destabilization of primary phagosomes [52] (ii) Mediate disruption of the double-membrane secondary phagosomes formed after cell-to-cell spread [53]
	*prfA*	PrfA (Pleiotropic regulatory factor)	Belongs to CAP (for catabolite gene activator protein)/FNR family, Dimeric protein and bindis binds to its DNA target sequences in dimeric form [54]
	*hly*	Listeriolysin O	(i) Mediates lysis of the primary phagosomes and required for the escape from the double-membrane vacuole [55] (ii) Regulates influx of calcium ions inside the host cell and plays a role in bacterium invasion [56] (iii) Downregulate the host immune system through dephosphorylation of H3 and deacetylation of H4 histones of the host cell and [57] (iv) Mitochondrial fragmentation. LLO-related toxins: ivanolysin (ILO) in *L. ivanovii* and ligerilysin (LSO) in *L. seeligeri* [58]
LIPI-2	*i-inlB2* *i-inlL* *i-inlK* *i-inlB1* *i-inlJ* *i-inlI* *i-inlH* *i-inlG* *smcL* *i-inlF* *i-inlE* *surF3*	Inernalins and SMase	(i) SMase disrupts phagosomes [59] (ii) Internalins mediate invasion into human epithelial cells [59]
LIPI-3	*llsAGHXBYDP*	Listeriolysin S	Haemolysin that is post-translationally modified and belongs to a family of modified virulence peptides, including streptolysin S and several as-yet uncharacterized members of the same family in other pathogens. LLS demonstrated to play a role in the survival of *L. monocytogenes* in PMNs and also contributes to its virulence in the mouse model [60]
LIPI-4	*lm4b_02324*	Maltose-6′-P-glucosidase	Putative 6-phospho-beta-glucosidase [61]
	*lm4b_02325*	Transcriptional antiterminator	Putative transcription antiterminator BglG family [61]
	*lm4b_02326*	Uncharacterized protein associated to PTS systems	Unknown [61]
	*lm4b_02327*	Membrane permease EIIA	Putative PTS system, cellobiose-specific enzyme component [61]
	*lm4b_02328*	Membrane permease EIIB	Putative PTS system, cellobiose-specific enzyme component [61]
	*lm4b_02329*	Membrane permease EIIC	Putative PTS system, cellobiose-specific enzyme component [61]

**Table 2 microorganisms-09-01856-t002:** Criteria used to define persistence of *Listeria monocytogenes* in literature.

Persistence Definition	Time Frame	Sample Number	Persisters Identified	Reference
PFGE type detected repeatedly for longer than 6 months	1 year			

		319	ST9, ST121	[64]


PFGE type isolated at least 3 occasions over the 16-month sampling period	16 months	2496	LS1, LS2, LS4, LS5, LS7, LS25, LS35, LS45	[65]
Genotypes isolated at least on 3 occasions with a minimum interval of 6 months between first and last isolation	3 years	1702	Serogroup IIa	[66]
PFGE type isolated at least six months apart	3 years	5869	P59, P6, P10, P32, P44	[67]
PFGE type isolated repeatedly, at least 4 times	2 years	1801	26 pulsotypes	[68]
CTs isolated at least 3 times with a minimum of 1 year between the first and last isolation	4 years	100	CT1526, CT1828, CT1833, CT1834, CT1836, CT1839	[69]

**Table 3 microorganisms-09-01856-t003:** Stress survival islands detected in *Listeria monocytogenes*.

Island	Acronym	Gene	Product	Function	Reference
Stress Survival Island-1	SSI-1	*lmo0444*	Hypothetical protein	Unknown	Ryan et al. [110]
		*lmo0445*	Transcriptional regulator	Regulation of transcription	
		*lmo0446 (pva)*	Penicillin acylase	Conversion of penicillin to 6-amino-penicillinate and phenylacetate—reduced susceptibility to penicillin V Survival in bile salts	Begley et al. [125]
		*lmo0447 (gadD1)*	Glutamate decarboxylase	Growth in mildly acidic pHs	Cotter et al. [126]
		*lmo0448 (gadT1)*	Amino acid antiporter		
Stress Survival Island-2	SSI-2	*lin0464*	Transcriptional factor	Involvement in alkaline and oxidative stress responses	Harter et al. [127]
		*lin0465*	Pfpl protease		

**Table 4 microorganisms-09-01856-t004:** Comparison of results obtained from crystal violet staining of biofilms in literature.

Method	Duration	Temp.	Substrate	Strains	Summary	Reference
MPA stained with crystal violet	24 h	37 °C and 10 °C	Bacterial biomass	19 persistent, 20 prevalent, 19 rare (27 genotypes)	Persistence/prevalence did not correspond to a higher biofilm formation, supplementation with NaCl in nutrient deprived cells improved biofilms, production ~5 times greater at 37 °C than at 10 °C	[159]
MPA/stainless steel sheets stained with crystal violet	24 h	37 °C	Bacterial biomass	ST1, ST2, ST5, ST8, ST9, ST87, ST121, ST199, ST321, ST388	All strains formed moderate biofilms on microtiter plates but not on stainless steel	[158]
MPA stained with crystal violet	16, 24 and 36 h	30 °C	Bacterial biomass	Range of serotypes from food/clinical samples	Lineage I isolates produced more biofilm, no significant difference in biofilm formation found between food/clinical isolates	[163]
MPA stained with crystal violet	224 h	37 °C	Bacterial biomass	Range of serotypes	Lineage II (persistent) isolates produced more biofilms, biofilm formation correlated with phylogenetic division but not serotype	[164]
MPA stained with crystal violet	24 h (37C) and 5 days (4C)	37 °C and 4 °C	Bacterial biomass	Range of serotypes from food/clinical samples	Higher biofilm formation at 37 °C with food isolates producing slightly more biofilm at both temperatures	[165]

**Table 5 microorganisms-09-01856-t005:** Genes involved in biofilm production by *L. monocytogenes*.

Gene	Product	Function	Reference
*degU*	Putative response regulator	*degU* is essential for flagellar synthesis and motility in *L. monocytogenes*. It is required for growth at high temperature, adherence to plastic surfaces and formation of efficient biofilms. It also functions in virulence of *L. monocytogenes*.	[168,170]
*flaA*	Flagellin A	Involved in initial attachment of *L. monocytogenes* in biofilm formation.	[169,171]
*agrBDCA*	Peptide-sensing system	Peptide-sensing system involved in quorum-sensing. Involved in early stages of biofilm formation. Also involved in virulence.	[172,173]
*bapL*	Putative cell wall-anchored protein	Surface adherence of *L. monocytogenes* in *L. monocytogenes*.	[174]
*sigB*	Major transcriptional regulator of stress response genes	Required to obtain wild-type levels of static and continuous-flow biofilms. Also involved in resistance of planktonic/biofilm cells to benzalkonium chloride and peracetic acid.	[175]
*dltABCD*	D-alanylation pathway	Further work needed to find specific function.	[177]
*phoPR*	Phosphate-sensing two component system	Further work needed to find specific function.	[177]

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
