# Peer review of "Application of Whole Genome Sequencing to Aid in Deciphering the Persistence Potential of Listeria monocytogenes in Food Production Environments"

_microorganisms, 2021, doi:10.3390/microorganisms9091856_

Round 1
Reviewer 1 Report
Abstract - it is appreciated that the authors identify that there is some hyperbole over the 'persister phenotype', as there is much information still required to define this phenotype.
Line 35 - this is a very clean, informative characterisation of the sensu stricto clade
Line 58 - if there is a reference on the directional transmission from facility to food, this would be helpful to support this theory (some do view it as theory not fact).
Line 77 - underlying factors of outbreak situations could be briefly described, such as scenarios where there is the wide distribution of contaminated foods and/or provision of such foods to vulnerable populations.
Figure 1 - This well worn Listeria diagram (adapted). Suggest that instead a novel (hopefully canonical) figure be drawn up on the stress survival features of Listeria, as is the focus of this review.
Section 3 - with the almost complete absence of references because the pathogenesis of Lm is well described and extensively reviewed elsewhere, suggest simply cite those reviews rather than devoting an entire page here.
Table 1 - The extensive findings associated with each function should not be ascribed to just three references - further original literature should be referenced. Possible role of ActA in biofilm formation not listed. The actA gene should not be capitalised
Line 153 - the hypervirulence theory has recently been reviewed elsewhere, and that article provided a detailed comparison on the nature of the lineages with different lifestyles of Lm (i.e., zoonotic vs saprophytic). Suggest that this theory is summarised and referenced here. See 10.1016/j.tim.2021.01.008
Line 183 - a good table to produce here would be a summary of different studies that claimed to reveal a persister phenotype, with the table to include the definitions and time frame (range) of their studies.
Line 190 - perhaps 'molecular epidemiological typing methods' or 'genomic'?
Line 239 - capitalise 'food' at the start of the sentence
Line 295 - what are the features of this plasmid? size, genes, etc.
Line 301 - 'and cadmium' not ', cadmium'
Section 9 - Agreed that biofilm characterisations of Listeria have been 'inconsistent'. Thus, another helpful field of literature to review/summarise here in a table would be to compare the methodological approaches and finding from select studies, describing the staining/microscopy method, duration, temperature, substrates, strains tested, and summary of observations.
Reviewer 2 Report
The paper could be reconsidered after major revision regarding updating of the reported informations.
Please see the attached file word for detailed comments

Round 2
Reviewer 1 Report
This is a much more focused and valuable review. I would just argue that the new title is not reflective of the review. You have reviewed phenotypic studies on biofilm, biocide and stress response much more extensively than the specific 'role of genome sequencing' in these areas.
Reviewer 2 Report
The authors did a big work in updating the manuscript and I found this new version is more scientifically sound.
Please see the attached file word for detailed comments
